# Mathematically aggregating experts' predictions of possible futures

A. M. Hanea[1], D. P. Wilkinson[1], M. McBride[2], A. Lyon[3], D. van Ravenzwaaij[4], F. Singleton Thorn[1], C. Gray[1], D. R. Mandel[5], A. Willcox[1], E. Gould[1], E. T. Smith[1], F. Mody[1], M. Bush[1], F. Fidler[1], H. Fraser[1], B. C. Wintle[1]

1 MetaMelb Lab, University of Melbourne, Melbourne, Victoria, Australia, 2 Centre for Environmental Policy, Imperial College London, London, United Kingdom, 3 DelphiCloud, Amsterdam, The Netherlands, 4 Faculty of Behavioural and Social Sciences, University of Groningen, Groningen, The Netherlands, 5 Cognimotive Consulting Inc., Toronto, Ontario, Canada

* anca.hanea@unimelb.edu.au

**Data Availability Statement:** All 3 datasets are available on OSF (https://osf.io/hkmv3/).

**Funding:** This research is based upon work funded by the Defence Advanced Research Projects

## Abstract

Structured protocols offer a transparent and systematic way to elicit and combine/aggregate, probabilistic predictions from multiple experts. These judgements can be aggregated behaviourally or mathematically to derive a final group prediction. Mathematical rules (e.g., weighted linear combinations of judgments) provide an objective approach to aggregation. The quality of this aggregation can be defined in terms of accuracy, calibration and informativeness. These measures can be used to compare different aggregation approaches and help decide on which aggregation produces the "best" final prediction. When experts' performance can be scored on similar questions ahead of time, these scores can be translated into performance-based weights, and a performance-based weighted aggregation can then be used. When this is not possible though, several other aggregation methods, informed by measurable proxies for good performance, can be formulated and compared. Here, we develop a suite of aggregation methods, informed by previous experience and the available literature. We differentially weight our experts' estimates by measures of reasoning, engagement, openness to changing their mind, informativeness, prior knowledge, and extremity, asymmetry or granularity of estimates. Next, we investigate the relative performance of these aggregation methods using three datasets. The main goal of this research is to explore how measures of knowledge and behaviour of individuals can be leveraged to produce a better performing combined group judgment. Although the accuracy, calibration, and informativeness of the majority of methods are very similar, a couple of the aggregation methods consistently distinguish themselves as among the best or worst. Moreover, the majority of methods outperform the usual benchmarks provided by the simple average or the median of estimates.

## 1 Introduction

Forecasting events or outcomes relies heavily on judgements elicited from people with expertise and knowledge in a relevant domain, whom we often call experts [1–3]. We ask experts to

Activity(DARPA), under Contract [HR001118S0047] as part of the SCORE program. The views and conclusions contained herein are those of the authors and should not be interpreted as necessarily representing the official policies, either expressed or implied, of DARPA, or the U.S. Government. The U.S. Government is authorized to reproduce and distribute reprints for governmental purposes notwithstanding any copyright annotation therein. The funders had no role in study design, data collection and analysis, decision to publish, or preparation of the manuscript. Cognimotive Consulting Inc. (DRM's affiliation) and DelphiCloud (AL's affiliation) did not provide any financial support and did not have any role in the study design, data collection and analysis, decision to publish, or preparation of the manuscript. The specific roles of the author affiliated with Cognimotive Consulting Inc. and DelphiCloud are articulated in the 'author contributions' section.

**Competing interests:** The University of Melbourne provided consultancy fees to Cognimotive Consulting Inc. in which DRM has a financial interest. However this did not alter our adherence to PLOS ONE policies on sharing data and materials because it did not interfere with the full and objective presentation of results and methods." The University of Melbourne provided consultancy fees to DelphiCloud in which AL has a financial interest. However this did not alter our adherence to PLOS ONE policies on sharing data and materials because it did not interfere with the full and objective presentation of results and methods.

predict what might happen under a set of circumstances, when we do not know what the future holds. In epidemiology, we might ask experts to forecast the prevalence of a disease in a population in five years' time, or when a given number of vaccine doses will be administered (e.g., see Good Judgment's Superforecasters). Intelligence analysts might seek predictions of the outcome of an important election, or the price of oil at some point in the future (e.g., [4]). Forecasting, and quantifying uncertainty through expert elicitation, may take two forms: eliciting probabilities, or eliciting values of a potentially measurable variable (like the future price of oil). Here, we are concerned with eliciting probabilities, which can be either thought of as a subjective degree of belief, or as a relative frequency.

Rather than rely on the subjective judgement of a single expert, it is typically considered best practice to elicit judgements from diverse groups [1, 5, 6], where group members can bring different perspectives, cross-examine each other's reasoning, and share information. When we have input from more than one individual, their judgements or forecasts need to be combined into a single estimate. They can either be combined behaviourally, where group members need to agree on a single judgement, or mathematically, where multiple judgements are aggregated using a mathematical rule, such as taking a simple group average (e.g., [7]).

Like any other type of data, expert judgements can be prone to errors and contextual biases. Since we use expert judgements in the same way we use empirical data, we should apply the same level of care and methodological rigor when eliciting and aggregating expert judgements. Methods for doing so are sometimes called *structured* protocols (e.g., [3, 8, 9]) and they aim to ensure that judgements are as reliable as possible, and are open to the same level of review and scrutiny as other forms of data.

Two of the three datasets analysed in this paper were elicited using a structured protocol called the IDEA protocol (e.g., [10]). The IDEA protocol encourages experts to Investigate, Discuss, and Estimate, and concludes with a mathematical Aggregation of judgements. For each event to be assessed, individuals research background information and investigate related sources of information; provide an anonymous estimate of the probability, together with their justifications for their estimates; receive feedback that reveals how their individual estimates differ from others' (a plot of all group members' estimates, together with their justifications); discuss differences in opinion and 'consider the opposite' (i.e., reasons why an event may or may not occur, or why a claim may or may not be true, as a group); and finally, provide a second anonymous estimate of the probability, incorporating insights gained through feedback and discussion.

To reflect imprecision around the elicited probability, we seek upper and lower bounds, in addition to a best estimate. When the probabilities can be interpreted as relative frequencies, the bounds can be interpreted as percentiles of the expert subjective probability distribution. When the relative frequency interpretation is not appropriate (i.e., when the probability of a unique event is elicited) the bounds may be criticised for lacking operational definitions (in a "classical" probabilistic framework). Many argue that the main reason to elicit bounds in such cases is to improve thinking about the best estimates [8, 9]. However, more research is needed to confirm these arguments (e.g., [11] and references therein).

The IDEA protocol results in two sets of probabilities for each event (prior and post group discussion and feedback). In this paper, an aggregation uses the final estimates (after discussion), unless otherwise specified. In a sense, the IDEA protocol benefits from both the wisdom of crowds (by aggregating more experts' estimates) and the "wisdom of the inner crowd" (e.g., [12]) by eliciting multiple estimates from the same individual. Although we do not average the multiple estimates of the same individual (instead, we combine estimates from multiple experts), we nevertheless consider the updated second estimate to be better than the first. When giving their second estimate, not only have the experts received (and had the chance to

challenge) feedback from their peers, but they have also sampled the evidence available to them multiple times, through the 3-step interval elicitation format [13], and across two rounds of elicitation.

There are many ways in which probability judgements can be mathematically aggregated (e.g., [3, 14]) and the IDEA protocol does not prescribe a certain mathematical aggregation. The most common method used in applications is the simple average, or an equally weighted aggregation method. However, evidence from cross-validation studies shows that performance-weighted aggregation methods (when more weight is assigned to people who have performed well in similar judgement tasks) lead to more calibrated, accurate and informative combined judgements, compared to equal-weighting of judgements [15–17]. To calculate differential weights for performance-weighted aggregations, seed (or calibration) data are needed. Seed data consists of experts' answers to seed questions, which are questions from the same domain as the target questions, for which the outcomes are known, or will become known, within the time frame of the study [18] (unlike the answers to the target questions). However, all the results and insights about performance-weighted aggregation methods' better performance (when compared to equally weighted aggregations) are obtained for elicitations of uncertain (theoretically) measurable, continuous quantities and their probability distributions (through a finite number of percentiles), rather than event probabilities.

In the current study, we focus on the elicitation of event probabilities, not the distribution of quantities. The number of seed questions needed to measure performance reliably, when probabilities are elicited is much larger than the number needed when eliciting quantiles of continuous quantities (e.g., [3]). This is a consequence of the instability of calibration and accuracy measures proposed for evaluating elicited probabilities [19]. To keep the elicitation burden manageable, and to reduce expert fatigue, seed questions are often avoided in probability elicitations. In this research we are interested in alternative ways of developing unequal weights, in the absence of seed questions. We propose using proxies for good performance to form weights. Such proxies are informed by previous research that investigated experts behaviour and particularities of their estimates that correlate with good performance. We differentially weight our experts' estimates by measures of reasoning, engagement, openness to changing their mind, extremity of estimates, informativeness, asymmetry of the elicited intervals surrounding best estimates, granularity of estimates, and prior knowledge. Using measures of experts' prior knowledge (or seed questions from adjacent domains, rather than the same domain as the target questions) is as close as we have to a seed dataset. Using such seed datasets to predict expert performance on the target questions is analogous to the transfer learning techniques from machine learning (e.g., [20]), but in a very data poor context and with much less sophisticated methods.

The vast majority of the aggregation methods that we outline in this paper are weighted linear combinations of experts' probability estimates (a.k.a. linear opinion pools), informed by our hypothesised proxies for good forecasting performance. The intuitive appeal, or the evidence behind these proxies is detailed when the methods are introduced. For comparison, we also consider the most popular aggregation methods used in the expert judgement literature—namely, simple averages of the experts' estimates or transformed estimates, and the median. Apart from linear opinion pools, Bayesian methods are sometimes proposed as mathematical aggregations of expert judgements. We propose two Bayesian aggregations as well. We measure the performance of these aggregation methods using three datasets. We introduce these datasets before the (aggregation) methods, since some of the aggregation methods are specific to one particular dataset, which is richer than the others. Nonetheless, future exercises for obtaining expert judgements may benefit from eliciting comparable additional information from the experts or participants, if it can be used to improve the quality of aggregated forecasts.

It is worth mentioning that the goal of this research is to understand human behaviour that predicts good performance so we only use real data. Simulation studies may prove useful for comparing aggregation methods, after we develop a better understanding of the limitations of the proposed aggregations. In this paper, we evaluate and compare each of the aggregation methods in terms of their accuracy, calibration and informativeness; with each of these measures reflecting different qualities of *good* judgement/performance.

The remainder of the paper is organised as follows: Section 2 details the specific measures of performance we are using to evaluate the different aggregations, Section 3 details the proposed aggregation methods, and the datasets they are compared on, and Section 4 shows these comparisons. The paper concludes with a discussion of findings, and the mathematical formulations of the aggregation methods can be found in S1 File.

## 2 Performance and scoring

When experts represent their uncertainty as a subjective probability, their assessments may then be scored. Roughly speaking, a scoring rule compares probabilistic forecasts against actual outcomes [21–23]. Despite the simplicity of this idea, there are many possible ways to score experts, each rewarding a different quality of the predictions. Some properties of the predictions need not be assessed relative to actual outcomes. We are concerned with scoring as a way of measuring those properties of expert subjective probability assessments that we value positively. Three properties that define expert performance will be discussed further: accuracy, calibration and informativeness.

### 2.1 Accuracy

Accuracy measures how close an expert's best estimate is to the true value or outcome. One widely used measure of accuracy is the Brier score [24]. The Brier score for events is the squared difference between an estimated probability (an expert's best estimate) and the actual outcome; hence it takes values between 0 and 1. Consider event or claim *c* with two possible outcomes *j*. The Brier score of expert *i* assessing event/claim *c* is calculated as follows:

$$BrierScore_{i,c} = \sum_{j=1}^{2} \left( p_{i,c,j} - x_{c,j} \right)^2,$$

where $p_{i,c,j}$ is expert *i*'s probability for event/claim *c* and outcome *j*, and $x_{c,j}$ is 1 if outcome *j* occurs and 0 otherwise. The above formula measures the accuracy of one estimate made by one expert about one event. Lower values are better (with zero representing perfect accuracy) and can be achieved if an expert assigns large probabilities to events that occur, or small probabilities to events that do not occur. An expert's accuracy can be then measured over many claims (*C* claims) and averaged to represent overall accuracy:

$$BrierScore_i = \frac{1}{C} \sum_{c=1}^{C} \sum_{j=1}^{2} (p_{c,j}^i - x_{c,j})^2 \tag{1}$$

The number of events and their overall sample distribution play an important role in interpreting such a score. By an overall sample distribution, we mean the inherent uncertainty of the events. This is also called the *base rate* and it is different (and often unknown) for each different set of events. However, its value, which has nothing to do with the expert's skill, contributes to the value of the average Brier score. This challenges the comparison of experts' scores calculated for different sets of events (e.g., [25]). Nevertheless comparisons will be meaningful when made on the same set of events.

Another measure of accuracy used for binary events is area under the curve ($AUC$). To define $AUC$ we need to define the receiver operating characteristic ($ROC$) curve (e.g., [26]). The $ROC$ curve represents the diagnostic ability of a binary classifier (in our case an expert) and it is obtained by plotting the true positive rate against the false positive rate. The best possible predictor yields a point in the upper left corner, the (0, 1) point of the $ROC$ space, representing no false negatives and no false positives. A random guess (e.g. the flip of a coin) would give a point along a diagonal line. The $AUC$ ranges from 0 to 1. An expert $i$ whose predictions are 100% wrong has an $AUC_i$ of 0; one whose predictions are 100% correct has an $AUC_i$ of 1 (e.g. [27]). $AUC$ is threshold independent because it considers all possible thresholds.

## 2.2 Calibration

Calibration compares the probabilities predicted by experts to the empirical probabilities. For example, if we group all instances where a 0.8 probability of events' occurrence was forecasted, we obtain a perfect calibration only if four out of five events occurred after such forecasts were issued. Before formally discussing one (of the many) calibration scores, let us introduce some necessary notation. Assume the experts are asked to assign events to probability bins of the following form $Bin_1 = (0.1, 0.9)$, $Bin_2 = (0.2, 0.8)$, $Bin_3 = (0.3, 0.7)$, etc. where the first number corresponds to the probability of the event occurring and the second number is the probability of the event not occurring (the probability of the complement). An expert would assign an event to $Bin_2$ if their best guess about the probability of that event's occurrence is 0.2. Let $p_k$ be the probability of occurrence that corresponds to bin $Bin_k$. Each expert assigns events to the different bins. Let $n_k$ denote the number of events assigned (by an expert) to the bin $Bin_k$, where $k$ takes integer values between 1 and 10, corresponding to the ten probability bins. Let $s_k$ denote the proportion of these events that actually occur; $s_k$ can be thought of as the empirical distribution of $Bin_k$, whose theoretical distribution is $p_k$. Ideally $s_k$ and $p_k$ should coincide. Nevertheless, in practice, they often do not.

The calibration is essentially a comparison between the empirical and theoretical distributions, per bin, per expert. For $c_k$ independent events/claims whose probability of occurrence is $p_k$ we can measure calibration through the average Brier score discussed in Section 2.1. The average Brier score can be decomposed into two additive components called *calibration* and *refinement* [28]. The calibration term, which is the component we are interested in, for a total of $C$ events/claims can be calculated as follows:

$$\sum_{k=1}^{10} \frac{c_k(p_k - s_k)^2}{C} \tag{2}$$

Very roughly, the refinement term (not precisely defined, nor used in this research) is an aggregation of the resolution and the inherent uncertainty of the events assessed. The resolution term measures the distance between the empirical distributions of the probability bins and the base rate.

## 2.3 Informativeness

Measuring the informativeness of experts' predictions does not require the actual outcomes. That is to say that an informative expert may well be poorly calibrated and/or not accurate. Hence, high informativeness is a desirable property only in conjunction with good calibration, or accuracy.

Experts' informativeness may be measured with respect to their choice of the probability bins. The choice (alone) of a more extreme probability bin (i.e., $Bin_1$ or $Bin_{10}$, which

correspond to probabilities close to 0 or 1) can give an indication of the expert's informativeness. The average *response* informativeness is measured as the average discrepancy between the expert's choice of probability bins and $Bin_5$, which corresponds to the (0.5, 0.5) uniform (least informative) distribution. The discrepancy is measured using the Kullback–Leibler divergence [29], also called the relative information of one distribution with respect to another. The response informativeness is defined in [30] as:

$$Info = \frac{1}{C} \sum_{i=k}^{10} c_k I(p_k, 0.5) \tag{3}$$

where $I(p_k, 0.5)$ is the Kullback–Leibler divergence of $p_k$ from 0.5. The response informativeness attains its minimum at zero, when all the variables are placed in the (0.5, 0.5) bin. A higher informativeness score is preferred since it indicates that more variables were placed in more extreme bins.

Some of the above measures assume that experts have placed events/claims in probability bins. However, most elicitation protocols ask experts to provide a best guess and an uncertainty interval around their best guess (e.g. [31]). The width of these intervals is a measure of the experts' confidence (sometimes called precision), or lack thereof, and can be an indication of how appropriate the size of the probability bins is. The width of the interval is sometimes used as a measure of informativeness as well (e.g., [32]). However, in this research we do not use this measure to evaluate the quality of predictions. Instead, we use it to form weights for some of the proposed aggregation methods.

## 2.4 Performance of aggregated predictions

The measures discussed in this section can indicate probabilistic prediction quality and serve multiple purposes. If they are applied to individual experts' predictions, they can be used to form weights which will then be used to construct a differentially weighted linear combination of judgements. This mathematically aggregated judgement can be thought of as a virtual expert whose prediction incorporates all experts' judgements, weighted according to their validity (as measured by prior performance). These aggregated judgements (virtual experts) represent the group's predictions and can be scored in the same way as experts' judgements. The measures described above will be used for comparing different mathematically aggregated judgements, rather than as a reward system for individual experts. The goal of constructing different aggregations and scoring them is to find the one that performs the best (according to one or more measures). This will further inform research into proxies for good performance, in those situations where performance cannot be measured as part of an expert elicitation due to various limitations.

## 3 Mathematical aggregation of elicited estimates

We present a suite of methods inspired by the available literature and current research. Because this research was motivated by the repliCATS project mentioned in Section 1, some of the aggregations we propose are specific to the information elicited in this project. So, before we launch into a description of our aggregation methods, we will first introduce the datasets on which they will be evaluated. The anonymised datasets are available on the OSF project page RepliCATS aggregation methods.

### 3.1 Datasets

Apart from the data collected for the repliCATS project (DARPA/SCORE), hereafter referred to as the **repliCATS dataset**, and described in Section 3.1.1, two other datasets informed this

research. These were collected by two separate research teams during a program funded by the US Intelligence Advanced Research Projects Activity (IARPA) and are described in Section 3.1.2.

**3.1.1 DARPA/SCORE.** The SCORE program (Systematizing Confidence in Open Research and Evidence) is funded by DARPA (Defense Advanced Research Projects Agency) in the US, and is one of the largest replication projects in history. It aims to develop tools to assign "confidence scores" to research results and claims from the social and behavioural sciences. Our team contributes to the this program, through the repliCATS project which uses a structured iterative approach for collecting participants' evaluations of the replicability of findings (a.k.a. claims) from the social and behavioural sciences. Specifically, we ask participants to estimate the "probability that direct replications of a study would find a statistically significant effect in the same direction as the original claim".

As part of this project, we conducted an experiment to explore how well IDEA groups (groups of participants using the IDEA protocol to estimate the probability of events/claims) performed when evaluating replicability using a set of "known-outcome" claims. These are social and behavioural science claims that have already been subject to a replication study and can be validated. These replication studies came from previous large scale replication projects, i.e., from Many labs 1, 2 or 3 [33–35] the Social Sciences Replication Project [36] or the original Reproducibility Project Psychology [37].

Data for this ***repliCATS dataset*** was collected using the IDEA protocol at a 2-day workshop in the Netherlands, in July 2019. The data collection was approved as part of the larger repliCATS project by the ethics board at the University of Melbourne (Ethics ID: 1853445). All participants signed consent forms prior to data collection. Participants were predominantly postgraduate students and early career researchers in psychology and behavioural research, with an interest in open science and metaresearch. Forecasts were validated against the outcome of the previous, high-powered replication study.

For each of the 25 claims assessed, participants were asked to provide (in addition to the probabilistic estimates) a comprehensibility rating and reasoning to support their quantitative estimates (see [38] for more detail). Prior to the workshop, participants also completed a quiz containing items relevant to evaluating replicability of research claims (statistical concepts and meta-research). The quiz was not compulsory and it was designed to cover subjects we expect the participants to be familiar with in order to reliably answer the target questions. Unlike calibration questions, the quiz questions may cover substantive and adaptive expertise, but less so normative expertise. However, quiz responses are the closest we have to a seed dataset, on which we can formulate performance weights.

The participants used the repliCATS platform [39] to answer all the questions and to record the accompanying reasoning. This allowed us to collect an extensive set of comments and reasons, which we then used to construct measures of reasoning breadth and engagement.

**3.1.2 IARPA/ACE.** The ACE program (Aggregative Contingent Estimation) was a forecasting "tournament" (2011–2015) funded by IARPA, aimed at improving the accuracy, precision, and timeliness of intelligence forecasts. The program engaged five university-based research teams to develop a range of best practice protocols for eliciting and aggregating accurate probabilistic forecasts. Teams deployed these tools to predict the outcomes of hundreds of real geopolitical, economic and military events that resolved, one way or another, typically within 12 months. An example question was "Will the Turkish government release imprisoned Kurdish rebel leader Abdullah Ocalan before 1 April 2013?". The near-future resolution dates allowed IARPA to validate the accuracy of the forecasts. Throughout the program, thousands of forecasters made over a million forecasts on hundreds of questions [40, 41].

Researchers from the University of Melbourne were part of one of the teams participating in the ACE program. This team elicited forecasts from IDEA groups, initially via email, and then through an online platform. Participants were asked to evaluate the questions, provide uncertainty judgements and justifications, and share materials and resources. Participants were also encouraged to rate and comment on the quality of the information shared by others. In the current paper, we refer to this as the ***ACE-IDEA dataset***. Participants' domain-relevant expertise ranged from self-taught individuals to intelligence analysts. The number of IDEA groups varied from 4 to 10 throughout the years, with each containing approximately 10 participants (but not all individuals answered all questions posed to their group). From the third year *Super-groups* were formed, comprising the best performing participants from the previous year. Super-group participants were unaware that their group was unique.

Another team in the ACE program was The Good Judgment Project (GJP), who won the tournament [41]. Data collected by the GJP team is available from the official GJP ACE data repository. We extracted data provided by 4844 participants who predicted subsets of 304 events. Even though participants had the chance to revise their estimate more than once, we have only extracted final estimates. We have also ignored any existing participant groupings, to minimise assumptions about how groups were organised across the large dataset, and over time. Instead, to mimic ACE-IDEA database and number and sizes of the IDEA groups, we used random subsets of this data. For each question we randomly chose 10 assessments and generated 10 random groups, assessing the same questions, but with no more that 10 assessments per question. We analyse these random subsets of the data, hereafter called the ***ACE-GJP dataset***, and we present averaged results.

The GJP elicited point estimate probabilities only, i.e., no upper or lower bounds. Participants in the GJP teams received training on how to interact effectively as a group, but engaged without external oversight. The GJP team encouraged *think again* and *consider the opposite* style practices [6] similar to the IDEA protocol, but without following a strict elicitation protocol.

### 3.2 Aggregation methods

The proposed aggregation methods can be organised in three different groups, each of which contain several related proposals. We introduce the motivation for these proposals, but leave the details and exact mathematical formulations for the S1 File. Each aggregation method is given an ID which is an abbreviation of the mathematical operations used to calculate it. The intention is to keep the aggregation methods names as intuitive and self-explanatory as possible, even though sometimes this will result in somewhat unconventional naming.

Several individuals (experts or participants) will assess events or claims (hereafter, "claims") whose outcomes are coded as 1 if the claim is considered true, and 0 otherwise. For each claim $c$, each individual $i$ provides assessments that the claim in question is true or false, by estimating three probabilities: $L_{i,c}$, which is a lower bound; $U_{i,c}$, an upper bound and $B_{i,c}$ which corresponds to the best estimate for the probability given by individual $i$ for claim $c$. These estimates satisfy the following inequalities: $0 \leq L_{i,c} \leq B_{i,c} \leq U_{i,c} \leq 1$.

Each claim is assessed by more than one individual and we aggregate their probabilities to obtain a group probability, denoted $\hat{p}_c$. We will further denote $\hat{p}_c(Method\ ID)$ as the aggregated probability calculated using the aggregation method with a given *ID*. For example, the simple average (the arithmetic mean) aggregation on $N$ individuals' assessments for claim $c$ is:

$$\hat{p}_c(ArMean) = \frac{1}{N}\sum_{i=1}^{N}B_{i,c} \tag{4}$$

Taking the average of the best estimates is the simplest and the most popular aggregation method. However its disadvantages are well known (and discussed in Section 3.2.1). A few alternatives proposed in order to overcome these disadvantages include transforming the best estimates prior- or post-averaging, or taking the median of the estimates, instead of the mean. Another type of average, proposed in some situations, is the average of the distributions fitted on the three estimates per claim given by each participant (in two of the datasets). We consider the average-type aggregations as one group. The second group of aggregations proposed (the largest of the three), is formed by a set of unequally weighted linear combinations of best estimates. The aggregations from this group only differ in how the weights are constructed, namely by different potential proxies for good forecasting performance. The last group of proposed aggregations contains two Bayesian approaches.

**3.2.1 Transformed averages and the median.**   The simplest way to aggregate group estimates is to take the unweighted linear average (i.e., the simple average or arithmetic mean of the best estimates $B_{i,c}$ for each claim). The aggregate estimate for claim $c$ is therefore calculated using Eq (4).

The simple average (arithmetic mean) of point probability estimates has proven to outperform individual estimates in numerous contexts (e.g., [8, 14]) and it is often used as a benchmark for which to compare other aggregation techniques. However taking a simple average of the estimates has several disadvantages (discussed in e.g., [42]) among which the lack of informativeness (often probability averages converge towards 0.5) and the sensitivity to outliers. To overcome the latter, taking the median rather than the mean is recommended (e.g., [43]). Not being influenced by outliers is an advantage as long as the outliers are not in the direction of the true outcome. If they are, the median's performance will be worse than the mean. We use both the simple average and the median as benchmark aggregations, and we call them *ArMean* and *Median*, respectively.

Modelling probabilistic estimates often involves transforming the probabilities to a more convenient scale. Log odds are often used to model probabilities (e.g., see [44, 45] for examples in generalised linear models and state estimation algorithms), typically due to the advantages of mapping probabilities onto a scale where very small values are still differentiable, and well studied distributions (like the Normal distribution) can be assumed when modelling. If we think of the discrepancy between the true probabilities and the probabilities estimated by the experts as errors, the error distribution on a probability scale, will not be symmetric, given that the scale is strictly bounded at both ends. However, when transformed, this inconvenience disappears. Numerical inconsistencies that may occur when extreme values are predicted (as zero or one) are avoided by slightly modifying such values away from the extremes. In the literature on expert elicited probabilities, the average of the log odds transformed individual best estimates has outperformed other benchmarks [43, 46]. Other aggregations of odds (e.g., the geometric mean of the odds, or logarithmic opinion pools), and other transformation of probabilities (e.g., probit transformations) have been proposed and shown to theoretically improve on several aspects of aggregations. It is however unclear that more sophisticated methods (requiring more theoretical assumptions) outperform the simplest ones on real, diverse and often small datasets. We refer to the simple average of log odds aggregation as *LOArMean*.

Previous research showed that averages of well-calibrated elicited probabilities are underconfident and methods to overcome this disadvantage were proposed by several authors (e.g. [42, 46, 47]). These methods basically propose different ways to extremise the simple average or any other linear opinion pool by shifting the aggregated value closer to either one or zero. Extremising by shifting the aggregated value farther from the base rate (instead of zero or one) was proposed in [48]. While this is a fascinating idea theoretically, in practice, more often than not, the base rate of the elicited events is unknown. We use the simplest

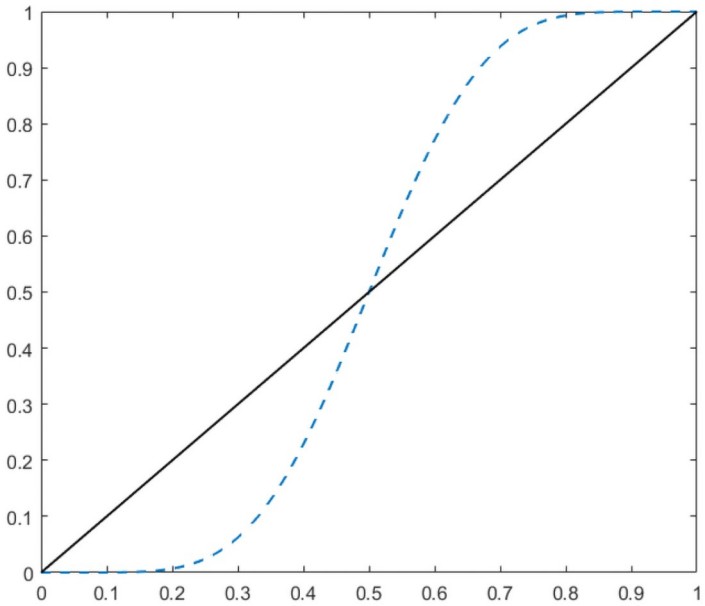

**Fig 1. Extremised probability scale.** The solid line represents the identity transformation (equivalent with a Beta(1, 1), which is the uniform distribution) and the dashed line represents the Beta(7, 7) transformation.

extremising technique initially proposed in [42], namely a beta-transformed arithmetic mean (called **BetaArMean**). This method takes the average of best estimates and transforms it using the cumulative distribution function (CDF) of a Beta distribution. The Beta distribution is parameterised by two parameters $\alpha$ and $\beta$, and in this analysis, we chose $\alpha$ equal to $\beta$ and larger than one. The justification for equal parameters is outlined in e.g., [46] and the references therein. The parameters for the Beta distributions used on the three datasets were optimised to maximise performance, which is impossible to do for datasets without known outcomes. However, we hypothesized that the Beta transformation will have a similar performance on datasets sharing the same characteristics (e.g., elicited with the same protocol, having similar groups sizes, asking for the probabilities of similar events). For example, the Beta parameters were equal to 7 for the repliCATS dataset, and the same value was used when aggregating predictions with unknown outcomes made throughout the project. Fig 1 shows how the values between 0 and 1 (solid line) are modified when transformed using the CDF of a Beta(7, 7), represented by the dashed line.

Another type of average, but this time of the distributions (constructed using all three estimates of participants per question) rather than of the best estimates alone, is proposed and called **DistribArMean**. This method assumes that the elicited best estimates and bounds can be considered to represent participants' subjective distributions associated with the elicited probability. That is to say that we considered that the lower bound of the individual per claim corresponds to the 5% percentile of their subjective distribution on the probability of replication, denoted $q_{5,i}$, the best estimate corresponds to the median $q_{50,i}$, and the upper bound corresponds to the 95% percentile, $q_{95,i}$. With these three percentiles, we build a minimally informative non-parametric distribution that spreads the mass uniformly between the three percentiles, such that the constructed distribution agrees with participant's assessments and makes no extra assumptions. This approach is inspired by methods for eliciting, constructing and aggregating distributions of continuous quantities, rather than probabilities [3].

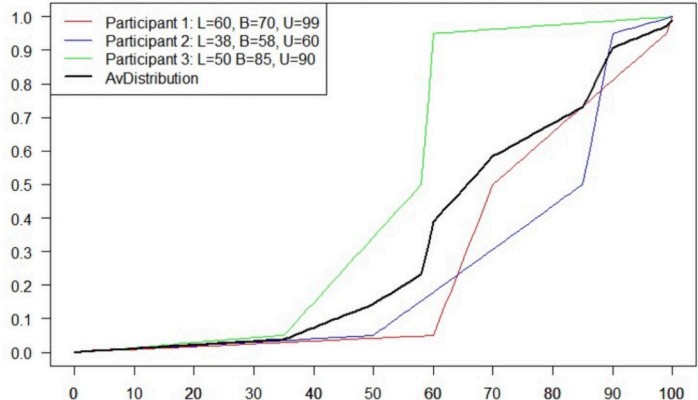

**Fig 2. The minimally informative non-parametric distributions associated with the estimates given by three participants for a particular claim and the average aggregated distribution.**

For example, assume three participants' estimates for a given claim are $\{q_{5,1} = 60\%, q_{50,1} = 70\%, q_{95,1} = 99\%\}$, $\{q_{5,2} = 38\%, q_{50,2} = 58\%, q_{95,2} = 60\%\}$, and $\{q_{5,3} = 50\%, q_{50,3} = 85\%, q_{95,3} = 90\%\}$. The three cumulative distribution functions built using these percentiles are shown in Fig 2. Their average is the aggregated distribution and the median of this aggregated distribution is the aggregated prediction.

**3.2.2 Weighted linear combinations of best estimates.** Even though weighted linear combinations of best estimates (of probability predictions) were shown to lack both calibration and informativeness even when the individual probabilities are calibrated (e.g., [42, 49]) these results are theoretical and were never strong enough to render weighted linear combinations useless in practice. Moreover, in practice, weighted linear combinations are more robust than sophisticated methods requiring more parameters' estimation (e.g., [50]).

The group of methods proposed in this section involves weighted linear combinations of individual assessments. All the weights are constructed based on properties of the participants' assessments, or on behaviours of the participants, which were either observed to correlate with good performance (in previous studies), or simply speculated to do so. The relative performance of these methods will be scrutinised and compared to gain insights into good proxies for good performance.

We denote the unnormalized weights by *w_method* (with subscripts denoting corresponding individuals or claims) and the normalised versions by *w̃_method*. All weights need to be normalised (i.e., to sum to one), but as the process is the same for all of them, in the S1 File, we will provide the formulae for the unnormalized weights. All differentially weighted combinations will take the form: $\hat{p}_c(Method\ ID) = \sum_{i=1}^{N} \tilde{w}\_method_{i,c} B_{i,c}$.

*IntWAgg* gives more weight to predictions accompanied by narrower intervals. The width of the interval (sometimes called *precision*, e.g., [51]) provided by individuals may be an indicator of certainty, and arguably of accuracy of the best estimate contained between the bounds of the interval. At least, this may be the case for quantitative judgements (rather than probabilities). For example, [52] found that experts provided narrower intervals with midpoints closer to the truth than novices (although both groups were overconfident). Similarly, [51] found that as the intervals provided by novices got narrower, the midpoints of those intervals were closer to the truth. There are many different ways to use interval width to weight the best estimates, with one possible approach being to weight according to the interval width across individuals for that claim. Previous studies that have weighted the best estimates or midpoints of

intervals by precision (interval width) have produced a more accurate aggregate than the simple average [51, 53]. However, these studies involved judgements where bounds were associated with a given level of confidence, and the true value was a quantity measured on a continuous scale rather than a binary outcome.

*IndIntWAgg* uses a re-scaled form of the individual interval width. Building on the previous method, we construct similar weights, but this time we account for the general tendency of an individual to be more or less confident (as expressed by the width of their intervals). Because of the variability in the widths of intervals participants give for different claims, we re-scale interval widths across all claims per individual, in this way rewarding certainty more if it is not their usual behaviour. We hypothesise that when a participant is more certain than usual, this certainty is informed by extra knowledge.

*VarIndIntWAgg* rewards larger variation in individuals' interval widths. Another speculation related to the ones above is that when a participant varies substantially in the given interval widths, this indicates a higher responsiveness to the existing supporting evidence to different claims. Such responsiveness might be predictive of more accurate assessors.

*AsymWAgg* rewards asymmetric intervals. Just as the width of an interval may be an indicator of knowledge and responsiveness to the existing supporting evidence, the asymmetry of an interval relative to the corresponding best estimate may likewise be an indicator. The implication is that participants who place their best estimate away from the middle of their uncertainty interval may have more thoughtfully considered the balance of evidence they entertained when producing their upper and lower bounds. However, this logic may be more appropriate for unbounded quantity judgements rather than for bounded probability judgements.

*IndIntAsymWAgg* combines the rewards for narrow intervals and asymmetry. The simplest way of achieving this is to multiply the previously defined and normalised weights.

*KitchSinkWAgg* uses weights that reward everything "but the kitchen sink". Building on all speculations discussed so far in this section, KitchSinkWAgg is an ad-hoc method developed and refined (but not yet publicly documented) using a single dataset (ACE-IDEA). This method is informed by the intuition that we want to reward narrow and asymmetric intervals, as well as variability between individuals' interval widths (across their estimates). However, the more desirable properties we add, the more parameters we have to estimate.

*DistLimitWAgg* rewards best estimates that are closer to the certainty limits. As mentioned when introducing informativeness, the choice of a more extreme best estimate may be a sign of confidence, hopefully driven by more knowledge on a particular subject. A preliminary analysis of the repliCATS dataset, showed a positive correlation ($r = 0.33$) between accuracy, as measured by the average Brier score, and distance from the nearest certainty limit, at the claim level. This indicates that better performance (small Brier scores) may be accompanied by shorter distances of best estimates to zero or one.

*ShiftWAgg* uses weights that are proportional with the change in estimates after discussion (with more emphasis on changes in the best estimates). An expert's openness to changing their mind after discussion is considered desirable and hence a proxy for potentially good performance. Calculating these weights is only possible when multiple rounds of judgements are elicited (like in the IDEA protocol). Previous analyses from [32] and [54] indicate that when participants change their second round judgement, often they become more accurate or more informative, or both. Therefore, weighting individuals' best estimates by the change in their estimates from the first round to the second round (after discussion) on a given claim may be beneficial.

*GranWAgg* rewards granularity of best estimates. Probability scales can be broken down into segments, with the level of segmentation reflecting granularity (i.e., $0.4 - 0.5$, versus $0.4 - 0.44, 0.45 - 0.49$, and so on) [55]. More skilled forecasters might be expected to have a finer

grained appreciation of uncertainty and thus make more granular forecasts. Accordingly, it may be sensible to give greater weight to participants who more frequently use "granular" estimates like 0.63 or 0.67 instead of rounded ones like 0.65 or 0.7. This is supported by the findings in [6] (who found that best performing forecasters made more granular forecasts than other forecasters) and [56].

*EngWAgg* uses individuals' verbosity to measure of engagement. When assessing claims, individuals have the chance to comment and engage in discussion with other participants. Previous studies showed that a high level of engagement may predict better accuracy of judgements [6, 9]. Therefore, this method gives greater weight to best estimates that are accompanied by longer comments.

*ReasonWAgg* rewards the breadth and diversity of reasons provided to support the individuals' estimates. When individuals provide multiple unique reasons in support of their judgment, this may indicate a breadth of thinking, understanding and knowledge about the claim and its context, and may also reflect engagement and conscientiousness. Qualitative statements made by individuals as they evaluate claims/studies were coded by the repliCATS Reasoning team, according to a detailed coding manual developed to ensure analysts were each coding for common units of meaning in the same sets of textual data. This manual emerged through an iterative process.

A preliminary analysis of the repliCATS dataset found a negative correlation between the participants' average Brier score and number of (unique) reasons they have flagged in the comments accompanying the numerical estimates. This means that participants who offer a larger number of distinct reasons to support their judgements are, on average, more accurate.

A variant of ReasonWAgg (called *ReasonWAgg2*) was formulated to incorporate not only the number of coded reasons listed by a given individual on a given claim, but also the diversity of reasons provided by a given individual across multiple claims. We refer the reader to the S1 File for details.

Even though most of the structured protocols for eliciting expert judgements encourage experts to provide reasons and rationales together with their numerical judgements, to the best of our knowledge, none formally model these qualitative datasets to inform mathematical aggregation of estimates. This is the first proposal of this kind, and much more qualitative data (and analysis) is needed to evaluate the extent of its advantages.

*QuizWAgg* uses quiz scores to calculate weights. As mentioned in Section 1, despite their usefulness (e.g., [16]), compulsory seed questions are often avoided, such that the elicitation burden is manageable. In the repliCATS project, we compromised by asking individuals to take on optional quiz before commencing the main task of evaluating research claims. Instead of a measure of prior performance on similar tasks, we can use a measure of participants' knowledge on a relevant domain, and how well they have understood the task or the question; these too might provide good indicators of their ability to make accurate predictions in that same domain. A separate analysis found a weak but statistically significant correlation between quiz scores and accuracy of prediction [57].

*CompWAgg* rewards higher (self-rated) comprehension levels. In the repliCATS project, before assessing a claim, individuals were asked to assess how well they understood it. A 7-point scale, where 1 corresponds to "I have no idea what it means" and 7 corresponds to "It is perfectly clear to me" is used for this comprehensibility question. Intuitively, the numerical estimates of the individuals who are confident they understood the claim may be weighted more. This is a speculation based on common sense, rather than supported by prior evidence.

**3.2.3 Bayesian aggregations.** Another type of mathematical aggregation proposed in the expert elicitation literature is Bayesian aggregation (e.g., [58–60]). Bayesian approaches treat the numerical judgements as data and seek to update a prior distribution using Bayesian methods.

This requires the analyst to develop priors and appropriate likelihood functions to represent the information implicit in the experts' statements. This is an incredibly hard and, in practice, a somewhat arbitrary task. Despite its theoretical appeal, in practice, Bayesian aggregation is much less used than opinion pooling. We propose two fairly simple, and fairly new Bayesian models which incorporate and take advantage of the particularities of the repliCATS dataset.

*BayTriVar* considers three kinds of variability around best estimates: generic claim variability, generic participant variability, and claim—participant specific uncertainty (operationalised by bounds). The model takes the log odds transformed individual best estimates as input (data), uses a normal likelihood function and derives a posterior distribution for the probability of replication. To complete the specification of the Bayesian model, normal and uniform priors need to be specified. The quantity of interest is the median of the posterior distribution of the mean estimated probability of replication. In Bayesian statistics the posterior distribution is proportional to the product of the likelihood and the prior and in this instance a Monte Carlo Markov Chain algorithm [61] is used to sample from this posterior distribution.

*BayPRIORsAgg* builds on the method above. The main difference between the two is that the parameters of the prior distributions are informed by the PRIORS model [62] which is a multilevel logistic regression model that predicts the probability of replication using attributes of the original study.

We summarise the aggregation methods and the proxies for good performance used to construct them (or the properties they represent) in Table 1. All these methods are implemented in the new **aggreCAT** R package [63].

## 4 Evaluations and comparisons

Before presenting and comparing results of the aggregation methods formulated in Section 3, a few properties of the datasets used in the analyses will be briefly discussed.

**Table 1. Aggregation methods.**

| Group | Method ID | Proxy/Property |
|---|---|---|
| Transformed averages and the median | ArMean | Wisdom of crowds |
| | Median | "Middle of the road" |
| | LOArMean | Skewed/Re-scaled wisdom of crowds |
| | BetaArMean | Extremised wisdom of crowds |
| | DistrArMean | Wisdom of crowds incorporating within expert uncertainty |
| Weighted linear combinations | IntWAgg | Precision |
| | IndIntWAgg | Precision relative to participants' behaviour |
| | VarIndIntWAgg | Variability in precision |
| | AsymWAgg | Considering the balance of evidence |
| | IndIntAsymWAgg | Precision & Considering the balance of evidence |
| | KitchSinkWAgg | All of the above (from this group) |
| | DistLimitWAgg | Extreme estimates |
| | ShiftWAgg | Openness to change their mind |
| | GranWAgg | Precision of point estimates |
| | EngWAgg | Engagement |
| | ReasonWAgg | Reasoning |
| | QuizWAgg | Prior knowledge |
| | CompWAgg | Comprehension |
| Bayesian aggregations | BayTriVar | Claim, Participant and Claim-Participant variability |
| | BayPRIORSAgg | Claim, Participant and Claim-Participant variability & Study characteristics |

**Table 2. Datasets characteristics.**

|  | repliCATS | ACE-IDEA | ACE-GJP |
|---|---|---|---|
| Number of participants | 25 | 150 | 4844 |
| Number of events/claims | 25 | 155 | 304 |
| Number of predictions per claim/event | 25 | between 6 and 48 | 10 |
| Extra information elicited | bounds, comprehensibility, reasoning, quiz | bounds | NA |
| Base rate | 0.52 | 0.19 | 0.26 |

Two of the datasets were obtained using the IDEA elicitation protocol described in Section 1, hence all estimates consist of lower bounds, upper bounds and best estimates for probabilities, elicited over two rounds. On the other hand, the third data set, namely the ACE-GJP dataset that we have extracted and used in this analysis contains point probability estimates only, equivalent to one round. Moreover the repliCATS dataset is much richer than the other two since it contains extra information on comprehensibility, engagement, reasoning, quiz performance, and prior information about the particular research that needs replication. A summary of what is elicited, from how many participants, for how many claims, and for what sort of claims (true or false) is presented in Table 2.

Apart from the differences summarised in Table 2, a few others are worth mentioning. The type of questions is different between the ACE datasets and the repliCATS dataset. Thinking of the probability of a study to be replicated involves different cognitive needs than thinking of the probability of a future sociopolitical event to occur. Not only the knowledge pool is different, but so is the time frame for the predictions.

## 4.1 Comparisons

All the differences discussed in the previous section will render any comparison of the aggregations' performance between datasets impossible. As a consequence we will only be signaling consistent relative behaviour of the aggregation methods, if any is observed. Moreover, we expect performance on the same dataset, but measured with a different measure of performance to vary as well. We will discuss these variations within and between datasets. [64] discuss these issues and propose a quantile metric to evaluate different aggregations under similar circumstances. In a sense, our approach of evaluating relative ranking is similar with the one proposed in [64].

Fig 3 shows the four scores discussed in Section 2, calculated for 15 of the proposed aggregation methods, for the three datasets described in Section 3.1. Because of the properties of the ACE-GJP dataset, only six of the 15 aggregations could be performed and scored. For the $AUC$ and the informativeness, higher scores are better, whereas for the average Brier score and the calibration, lower scores correspond to better performance.

We are interested in the relative rankings of the aggregation methods (from worst to best) and how consistent these are across datasets. At a first glance we can notice that the scores (irrespective of the measure) are not very different between the methods, when evaluated on the same dataset. The average Brier scores and the $AUC$ scores are the best examples of this "uniformity".

The average Brier scores range from 0.11 to 0.15 on the repliCATS dataset, from 0.09 to 0.1 on the ACE-IDEA dataset, and from 0.06 to 0.08 on the ACE-GJP dataset. While all these scores indicate good accuracy, differences of order $10^{-2}$ are hardly enough to justify strong orderings.

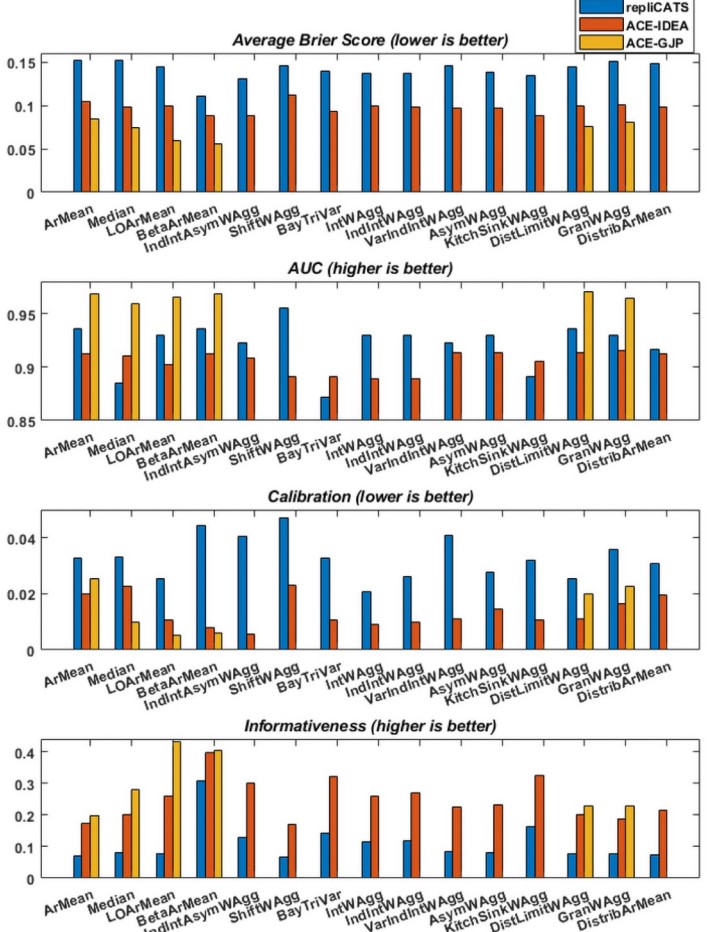

**Fig 3. The performance of 15 of the aggregation methods on the three datasets as measured by four performance measures.**

The same signal is observed when looking at the $AUC$ scores. These range (for all aggregations and datasets) from 0.85 to 0.97 which can all be considered very good scores. Unfortunately, no absolute threshold, or objective definition a "good" $AUC$ score exist; scores closest to one are better, but how much better is entirely application dependent. The $Y-axis$ range of the $AUC$ was cropped to make the differences more visible, but it is difficult to objectively interpret the magnitude of these differences.

The calibration measure is the most unreliable of these measures as it is very sensitive to the number of questions participants answer [19]. Notably, on this measure, BetaArMean has one of the worst scores on the repliCATS data set, but amongst the best on the other two datasets.

For the informativeness scores, one method clearly outperforms the rest on the repliCATS and ACE-IDEA datasets and comes second on the ACE-GJP dataset. This is the BetaArMean aggregation. For the ACE-GJP, BetaArMean is very close to the best (0.4), with the best being LOArMean (0.43). ArMean is amongst the methods with the lowest informativeness scores (on all dataset), which is not surprising. Informativeness measures departure from a 0.5 probability assessment, hence, it makes perfect sense for the BetaArMean aggregation, which is an extremising method to score amongst the most informative.

| Aggregation | repliCATS | | | | ACE-IDEA | | | | ACE-GJP | | | |
|---|---|---|---|---|---|---|---|---|---|---|---|---|
| | ABS | AUC | CAL | INFO | ABS | AUC | CAL | INFO | ABS | AUC | CAL | INFO |
| BetaArMean | 0.11 | 0.94 | 0.04 | 0.31 | 0.09 | 0.91 | 0.01 | 0.40 | 0.06 | 0.97 | 0.01 | 0.40 |
| IndIntAsymWAgg | 0.13 | 0.92 | 0.04 | 0.13 | 0.09 | 0.91 | 0.01 | 0.30 | | | | |
| KitchSinkWAgg | 0.13 | 0.89 | 0.03 | 0.16 | 0.09 | 0.91 | 0.01 | 0.33 | | | | |
| IndIntWAgg | 0.14 | 0.93 | 0.03 | 0.12 | 0.10 | 0.89 | 0.01 | 0.27 | | | | |
| IntWAgg | 0.14 | 0.93 | 0.02 | 0.12 | 0.10 | 0.89 | 0.01 | 0.26 | | | | |
| AsymWAgg | 0.14 | 0.93 | 0.03 | 0.08 | 0.10 | 0.91 | 0.01 | 0.23 | | | | |
| BayTriVar | 0.14 | 0.87 | 0.03 | 0.14 | 0.09 | 0.89 | 0.01 | 0.32 | | | | |
| DistLimitWAgg | 0.14 | 0.94 | 0.03 | 0.08 | 0.10 | 0.91 | 0.01 | 0.20 | 0.08 | 0.97 | 0.02 | 0.23 |
| LOArMean | 0.14 | 0.93 | 0.03 | 0.08 | 0.10 | 0.90 | 0.01 | 0.26 | 0.06 | 0.96 | 0.01 | 0.43 |
| ShiftWAgg | 0.15 | 0.96 | 0.05 | 0.07 | 0.11 | 0.89 | 0.02 | 0.17 | | | | |
| VarIndIntWAgg | 0.15 | 0.92 | 0.04 | 0.08 | 0.10 | 0.91 | 0.01 | 0.23 | | | | |
| DistribArMean | 0.15 | 0.92 | 0.03 | 0.07 | 0.10 | 0.91 | 0.02 | 0.22 | | | | |
| GranWAgg | 0.15 | 0.93 | 0.04 | 0.08 | 0.10 | 0.92 | 0.02 | 0.19 | 0.08 | 0.96 | 0.02 | 0.23 |
| ArMean | 0.15 | 0.94 | 0.03 | 0.07 | 0.10 | 0.91 | 0.02 | 0.17 | 0.08 | 0.97 | 0.03 | 0.20 |
| Median | 0.15 | 0.88 | 0.03 | 0.08 | 0.10 | 0.91 | 0.02 | 0.20 | 0.07 | 0.96 | 0.01 | 0.28 |

**Fig 4. Numerical scores measuring the performance of 15 of the aggregation methods on the three datasets.** The table is ordered from best to worst performance as measured by the average Brier score (ABS) on the repliCATS dataset.

As mentioned, none of the numerical differences are significant, but for the sake of completeness and a better visualisation of the aggregation methods' ranking we present the numerical results (rounded to two decimals) in Fig 4. A grey scale is used to indicate best to worst performance, with darker shades representing better performance.

BetaArMean's performance stands out as the best on at least three measures of performance on each of the datasets. On the other hand, ArMean's performance stands out as the worst on at least two of the measures, on each dataset. This indicates that most aggregation methods outperformed the standard benchmark represented by the simple average. The other sometimes used benchmark, the Median is also amongst the worst ranked methods on at least one measure, on each dataset. An aggregation whose performance is slightly puzzling is the ShiftWAgg which is the best performer as measured by the *AUC* on the repliCATS dataset and the amongst the worst on all the other measures and on the ACE-IDEA dataset.

As mentioned, the full set of aggregation methods can only be used on the repliCATS dataset. Fig 5 shows the same scores as above, calculated for all the aggregation methods described in Section 3, but only for the repliCATS dataset. Despite the redundancy of the information presented, Fig 5 allows a complete comparison between all methods, albeit on only one dataset. There are 21 different aggregation methods, with the two reasoning-weighted linear combinations having very similar formulations.

For each of the four scoring measures, the values are ordered from worst to best performance, and so the order of the aggregation methods changes per subplot accordingly. On the right hand side of Fig 5 we can observe the best performers and on the left hand side the worst ones. BetaArMean is one of the most accurate (both in terms of the *AUC* and the Brier score) and the most informative relative to the other methods, and so is KitchSinkWAgg if we only measure accuracy in terms of the Brier score. However the *AUC* for KitchSinkWAgg is amongst the worst. Apart from a good *AUC* score for ArMean, both ArMean and the Median are outperformed by the majority of methods. LOArMean has mediocre performance on all measures apart from the calibration (the most unstable measure), where it ranks amongst the best. IntWAgg and IndIntWAgg have almost equal performance on all measures, suggesting that re scaling the weights with respect to the largest interval does not do much to improve performance of the aggregation. However, four of the five best performing methods, in terms of the Brier score, do incorporate interval width in one form or another. VarIndIntWAgg does

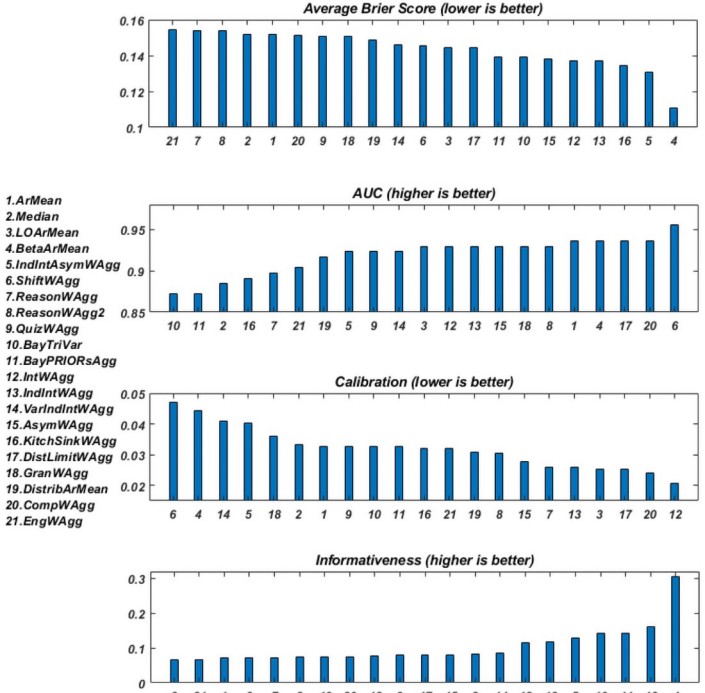

**Fig 5. The relative performance of the 21 aggregation methods on the repliCATS dataset.**

not outperform IntWAgg, suggesting that variability in the interval lengths do not tell us much in this dataset. IntWAgg and AsymWAgg have very similar performance and their combination produces slightly improved Brier score and informativeness, but worse calibration and *AUC*. We speculate that these differences are equivalent to sampling fluctuations rather than true signals. Many other datasets should be investigated and compared in order to increase the credibility of these signals.

The Bayesian methods produce equal scores on all measures, which may indicate that the influence of the prior distribution is very weak. Their accuracy as measured by the Brier score and their informativeness are among the best, but their *AUC* is the worst (though still larger than 0.85).

The reasoning-weighted methods have mediocre performance on all measures apart from the Brier score which indicates bad performance relative to the other methods. One of the possible reasons may be the setting of the data collection. The repliCATS data was collected via a face-to-face workshop where participants had the chance to discuss live with their fellow participants but they were encouraged to record their reasoning on a dedicated platform. The extent of their diligence in doing so is questionable.

## 5 Discussion

In this paper, we have explored the performance of 21 methods for aggregating probability judgements from experts. Although we were unable to compare every method on all three datasets, we could nonetheless detect consistent signals.

The beta-transformed arithmetic mean (BetaArMean) outperformed most of the other aggregations on all data sets. BetaArMean takes the average of best estimates and transforms it using the cumulative distribution function of a beta distribution, effectively *extremising* the

aggregate. This method also performed well in analyses by [46]. It is not surprising that this method was so successful in terms of informativeness (since more extreme estimates are closer to certainty limits). But the method also performed well on the other scoring measures, producing aggregates that proved to also be accurate and well calibrated. We caution against valuing extreme estimates based solely on informativeness, since this does not guarantee accuracy; and judgements that appear informative but turn out to be wrong are the most misleading. It stands to reason that an extremising method would improve accuracy on datasets where probability judgements are on the 'right' side of 0.5 the majority of the time (corresponding to high classification accuracy). If the average judgement is already reasonably accurate, the error will be greater when the aggregate is more moderate (closer to 0.5). For the datasets analysed here, the average judgements (the ArMean aggregation) had classification accuracy larger than 84%. This high classification accuracy was driven by a fairly high average participant classification accuracy (75% for the repliCATS dataset, 79% for the ACE-IDEA, and 78% for the ACE-GJP). In settings where the classification accuracy is low, an extremising methods will perform poorly. Anti-extremising methods were proposed in [48]. However, knowledge about specific base rates is essential to these methods.

A subset of methods that performed reasonably well relative to the others (on all datasets) are the weighted linear combinations where the weights are informed by the lengths or the adjustment of the intervals around the best estimates. Although inconclusive, these results lend some justification for eliciting intervals. In part, uncertainty bounds may reflect subjective confidence, which has acquired a bad reputation in psychology and other fields [65] through the abundance of *overconfidence* research that repeatedly reveals a mismatch between confidence and accuracy [66]. Yet, giving greater weight to more confident judgements on any given claim (i.e., flexibly adopting or weighting the judgement of one or another judge, without assuming the same person will always be the best judge, [65]), has shown merit in improving accuracy, but only in *kind* versus *wicked* environments [67]. In this case, a kind environment is one that is more predictable, where our knowledge of the world, or of the task, more often reflects reality than it is misguided.

When eliciting judgements from experts, other information could be requested alongside the focal prediction or estimate (aside from bounds). For example, recent research proposes eliciting both the point estimate of interest as well as asking the judge to predict the average estimate that would be given by others, and using this to partly inform weights [68]. This is underpinned by the idea that judges who have a good grasp of other individuals' opinions will be better at judging the question itself [69]. [68] suggest that this approach might be particularly useful in group contexts where participants are given the opportunity to share information and cross-examine each other's judgements (as when using the IDEA protocol). Future research could compare the usefulness of eliciting different types of additional information beyond the target judgement.

One method that had a constant mediocre performance on all datasets and measures was DistribArMean, which takes the mean of the non-parametric distributions. In order to construct and aggregate percentiles, rather than probabilities, it assumed that the lower bound given by the participants corresponded to the 5% percentile of their subjective distribution on the probability of an outcome, the best estimate corresponded to the median, and the upper bound corresponded to the 95% percentile. This is quite a strong and arguably arbitrary assumption, especially as experts were not told to give bounds that corresponded to any particular percentile. Even if participants were asked to think of bounds as percentiles of subjective distributions, but these percentiles were not fixed (to the 5% and 95%), these bounds would not represent the same percentiles for all experts or all claims. We conjecture that this faulty

assumption is a possible contributing factor for the unremarkable performance of this aggregation method.

The ArMean and the Median aggregations ranked amongst the worst performing methods consistently. Outperforming these two benchmark methods suggests that the hypothesised proxies for good performance may indeed improve aggregated predictions. The poor performance of the Median lends some support to the idea that outliers may contain valuable information; that is, outliers may be in the direction of the *right* answer. These results are somewhat in contradiction to results presented in [64], and even with the performance of the median in the ACE-GJP dataset (which is slightly better than the that of the mean). In both cases, the elicitation did not benefit from the settings, hence the advantages of the IDEA protocol. When the IDEA protocol is employed, the participants have the opportunity to change their estimates in light of discussion. If they choose to stay with their original estimates (rather than move in the direction of the majority), they may have good reasons for doing so.

Do these results support our hypotheses that weighting by proxies for good performance is worthwhile? It is not clear from our results that this is the case. From these results alone, it may be difficult to justify the additional elicitation burden of gathering additional information from participants (e.g., asking them to complete a quiz). But since a large subset of proxy performance-weighted methods could only be applied in a single, relatively small dataset, caution should be employed before these findings are generalised beyond this specific case. This demonstrates, once more, the importance of gathering more calibration data to further explore this question.

Similarly, we cannot conclude much about the Bayesian methods, which displayed contradicting performance when measured on the $AUC$ versus the Brier score. Their informativeness is attractive but not on its own. Nonetheless, the Bayesian aggregation methods are interesting as a proof of concept, and also as a reminder that although incorporating prior information may be helpful, it may be insufficient or misleading. Prior information is however crucial when formulating performance based weights, or informing parameters of the BetaArMean.

From a decision-theoretic perspective, the tension between the accuracy, the calibration, and the informativeness of predictions poses an important challenge for evaluating elicited and aggregated forecasts. Decisions about the most appropriate aggregation method should consider that high informativeness should only be valued subject to good calibration, or good accuracy. The choice between these two though can be further informed by the end-user values.

None of the results of our analysis can be considered definitive or strongly conclusive; while some corroborated pre-existing findings, others countered the evidence of published studies. However, the main contribution of this study is to explore proxies for good predictive performance informed by human judgment behaviour and prior knowledge, and to build this information into better performing aggregation methods. Although we detected promising signals, further research is needed for stronger conclusions.

## Supporting information

**S1 File.**
(PDF)

## Author Contributions

**Conceptualization:** A. M. Hanea, D. P. Wilkinson, F. Singleton Thorn, A. Willcox, E. Gould, E. T. Smith, F. Mody, M. Bush, F. Fidler, H. Fraser, B. C. Wintle.

**Data curation:** A. M. Hanea, D. P. Wilkinson, F. Singleton Thorn, C. Gray, A. Willcox, E. Gould, E. T. Smith, F. Mody, M. Bush.

**Formal analysis:** A. M. Hanea, E. T. Smith, F. Mody.

**Methodology:** A. M. Hanea, M. McBride, A. Lyon, D. van Ravenzwaaij, F. Singleton Thorn, D. R. Mandel, E. Gould, E. T. Smith, M. Bush, F. Fidler, H. Fraser, B. C. Wintle.

**Software:** D. P. Wilkinson, C. Gray, A. Willcox, E. Gould.

**Supervision:** F. Fidler.

**Writing – original draft:** A. M. Hanea, B. C. Wintle.

**Writing – review & editing:** A. M. Hanea, D. P. Wilkinson, M. McBride, D. van Ravenzwaaij, F. Singleton Thorn, D. R. Mandel, E. Gould, E. T. Smith, M. Bush, H. Fraser, B. C. Wintle.

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
