## [Decision Letter · Decision Letter 0]

4 Mar 2021

PONE-D-21-01369

Mathematically aggregating experts’ predictions of possible futures

PLOS ONE

Dear Dr. Hanea,

Thank you for submitting your manuscript to PLOS ONE. After careful consideration, we feel that it has merit but does not fully meet PLOS ONE’s publication criteria as it currently stands. Therefore, we invite you to submit a revised version of the manuscript that addresses the points raised during the review process.

In particular:

Improve presentation of aggregation rules (avoid "laundry list"), providing rationale, intuition behind each and evidence from previous literature that such a scheme could be reasonable.When presenting results, highlight the main points of the story.Try to establish links with similar concepts in related disciplines (e.g. computer science / AI / statistics).Make clear what this work adds to the literature.Address the many constructive comments by the reviewers.

We look forward to receiving your revised manuscript.

Kind regards,

Luca Citi, PhD

Academic Editor

PLOS ONE

Journal Requirements:

2)  Thank you for stating the following in the Competing Interests section:

[The authors have declared that no competing interests exist.].   

We note that one or more of the authors are employed by a commercial company: DelphiCloud and Cognimotive Consulting Inc.

i. Please provide an amended Funding Statement declaring this commercial affiliation, as well as a statement regarding the Role of Funders in your study. If the funding organization did not play a role in the study design, data collection and analysis, decision to publish, or preparation of the manuscript and only provided financial support in the form of authors' salaries and/or research materials, please review your statements relating to the author contributions, and ensure you have specifically and accurately indicated the role(s) that these authors had in your study. You can update author roles in the Author Contributions section of the online submission form.

ii. Please also provide an updated Competing Interests Statement declaring this commercial affiliation along with any other relevant declarations relating to employment, consultancy, patents, products in development, or marketed products, etc. 

3) We note that you have stated that you will provide repository information for your data at acceptance. Should your manuscript be accepted for publication, we will hold it until you provide the relevant accession numbers or DOIs necessary to access your data. If you wish to make changes to your Data Availability statement, please describe these changes in your cover letter and we will update your Data Availability statement to reflect the information you provide.

Reviewers' comments:

Reviewer's Responses to Questions

**Comments to the Author**

1. Is the manuscript technically sound, and do the data support the conclusions?

Reviewer #1: Yes

Reviewer #2: Yes

Reviewer #3: Yes

Reviewer #4: Yes

2. Has the statistical analysis been performed appropriately and rigorously? 

Reviewer #1: No

Reviewer #2: Yes

Reviewer #3: No

Reviewer #4: I Don't Know

3. Have the authors made all data underlying the findings in their manuscript fully available?

Reviewer #1: Yes

Reviewer #2: Yes

Reviewer #3: No

Reviewer #4: No

4. Is the manuscript presented in an intelligible fashion and written in standard English?

Reviewer #1: Yes

Reviewer #2: Yes

Reviewer #3: Yes

Reviewer #4: No

5. Review Comments to the Author

Reviewer #1: This is a very nice work, I like reading it very much. The way it is written is really clear, and using an astounding English writing that I am unable to judge. I found the paper very informative in terms of a "Review" of the different aggregation methods that can be used from experts' predictions.

I can perform two main criticism of this work. The first one is that, though it may be influenced from my background, I think the manuscript can benefit if it is more connected to Computer Science concepts. For instance, I found that the provided definition for Entropy is not the standard definition given in computer science or statistical sciences.

The second criticism is that the evaluation of the 22 methods lack any statistical proof about the obtained results. I am not pointing towards some extreme statistical hypothesis test but at least some paragraph stating what kind of control experiments authors executed to verify that obtained results are not affected by some uncovered bias. For instance, what happens if Authors run the 22 tests on an artificial dataset that is produced by an agent that randomly produces responses ? I do not think that per se that invalidates anything about this work, but anyway, it would beneficial to understand if the results that Authors finally obtained are affected by chance or not.

Some minor issues that I found on the manuscript:

43-45: Are Authors suggesting that when the relative frequency interpretation is not appropriate, the reason why it should be encouraged to elicit bounds while making a probabilistic forecasting decision is because it helps the expert to elucidate a best estimate B i,c ?

104: I suggest authors that they may link the idea of analyzing or considered prior performance in a similar domain, connected with the concept of transfer learning in Computer Science/Machine Learning.

157: Equation (1): I would like to suggest Authors to use shorter variable names in Equations, or greek symbols. The mathematical notation using very long variable names makes them more difficult to read.

132: the important concept of calibration is used here, before being explained in detail in section 2.1.2

166: I suggest authors to provide some reference to a prior work on ROC curve definitions.

177: In this sentence, I think the word "Calibration" could be replaced by "Accuracy" and it will be valid. The explanation provided for calibration later in this paragraph is excellent, but I found that sentence a little bit confusing. Is "Calibration" the same thing as "Accuracy" ?

199: Where is the "Refinement" term ?

202: Entropy is used here, very roughly defined, but it is defined in more detail later in Section 2.1.3.

Section 2.1.3: The definition of Entropy in Equation (3) is a little bit confusing. I think it is a very important concept that should be defined more properly (in terms of having a more self-contained manuscript). I would suggest to cite and link with this article "A modified belief entropy in Dempster-Shafer framework, Zhou 2017" and provide the Shannon Entropy definition used in that article (which you could later modify to this particular problem of having the entropy of the particular distribution (pk, 1-pk).

Section 3: The idea of seed questions could be linked to Transfer Learning (as mentioned earlier) as well as with Supervised Learning.

297: I suggest Authors to add Bi,c to "simply takes the average of the best estimates B i,c of each individual)" for clarity.

301: Why in this line "the mean" (i.e. I assume arithmetic mean) is different from the "The simple Average.." in line 299.

318: The concept of "extremizes the mean estimate" I believe that it could be further defined and clarified (it is used again in line 692).

358: I think the name "KitchSinkWAgg" is somehow unexpected in this context, and I think it could be further clarified (I was totally curious about that). Additionally and more broadly, I think the manuscript is going to be improved if an extra table with all the different methods are summarized and described (what aspect of the proxies are each one of them trying to emphasize).

438: I think the NVivo software could be referenced (if there is any reference), or at least clarified something else about it.

527: There is a broken reference in the manuscript.

658: I am not sure what exactly the pronoun "Its" is referring to in this sentence.

Figure 2: I think you have 22 methods, not 21.

One last word regarding the Discussion section, it was hard for me to state clearly what are the findings of this work. Perhaps, stating the findings, one by one, in a clear manner helps readers to identify them.

Reviewer #2: This study reports a pure experimental analysis on existing methods of information aggregation. 22 aggregation methods are examined with several data sets. The verifications rely on three criteria including accuracy, calibration, and informativeness. The evaluation results are clear to me: the beta-transformed arithmetic mean is the best, and Arithmetic mean of the non-parametric distribution is the worst. The statement is clear and complete. I enjoyed reading this paper. The present results are convincing to me.

Reviewer #3: The authors consider the problem of aggregating experts' probability predictions of a future binary outcome. They consider many different aggregators that derive from the weighted average and differ in the way the weights are calculated. They apply the aggregators to three real-world datasets and compare the performances of the aggregators in terms of several well-established criteria. Even though no single aggregator emerges as a clear winner, the beta-transformed arithmetic mean (BetaArMean) performs very well.

Overall, I enjoyed reading the paper, and I believe the literature would benefit from an extensive comparison of different weighting schemes. However, the paper needs work before being published. In the attached document, I have listed both general and specific comments. I hope they help the authors to improve this paper.

Reviewer #4: Review of PONE-D-21 Mathematically aggregating experts’ predictions of possible futures

I was hoping to gain some new insights into aggregation when I read this paper. I wish I had. This paper seems more like a report to a government agency than a journal article that tells a story about how to improve aggregation. Many of the methods considered seem way too similar to matter, and yet they are considered anyway. The writing is difficult to follow and the graphs are too small to show the differences the authors claim they find – the best and worst aggregation rules across datasets. I don’t see a good explanation for WHY these aggregation rules are the best and worst. I don’t understand what it was about the data sets that may had produced the differences, and I’m just not learning enough from the paper.

Perhaps I could be convinced that a very different version of this paper could be published. The new version would have to get right to the point and focus on the key points – not a laundry list of aggregation rules. It would also have to include an explanation of why the results came out the way they did. Presentation of the arguments is not strong in the current form. Figures 1 and 2 compare too many rules, and they don’t really highlight the main points of the story. Sorry to be negative, but I think the authors should focus more on what they add to the literature that is different, new and will result in progress on these issues.

6. PLOS authors have the option to publish the peer review history of their article (what does this mean?). If published, this will include your full peer review and any attached files.

Reviewer #1: No

Reviewer #2: No

Reviewer #3: **Yes: **Ville Satopää

Reviewer #4: No

---

## [Author Response · Author response to Decision Letter 0]

8 Jun 2021

Please see the attached document Answers2Comments.docx for detailed answers to all comments.

---

## [Decision Letter · Decision Letter 1]

5 Jul 2021

PONE-D-21-01369R1

Mathematically aggregating experts’ predictions of possible futures

PLOS ONE

Dear Dr. Hanea,

Thank you for submitting your manuscript to PLOS ONE. After careful consideration, all three reviewers and the editor agree that the paper has improved noticeably through the revision and that is practically ready for publication. However, there are a few remaining minor points raised by the reviewers that we would like the authors to consider for their final version. To speed up the process and also in consideration of the reviewers' time, your new submission will not be sent for another round of reviews but will be evaluated by the editor only.

A brief rebuttal letter that quickly responds to the last few outstanding points.In the interest of time, there is no need to submit the marked-up copy of your manuscript (if the system requires a file to be uploaded, feel free to submit an empty document labeled 'Revised Manuscript with Track Changes').An unmarked version of your revised paper without tracked changes. You should upload this as a separate file labeled 'Manuscript'.

We look forward to receiving your revised manuscript.

Kind regards,

Luca Citi, PhD

Academic Editor

PLOS ONE

Journal Requirements:

Reviewers' comments:

Reviewer's Responses to Questions

**Comments to the Author**

1. If the authors have adequately addressed your comments raised in a previous round of review and you feel that this manuscript is now acceptable for publication, you may indicate that here to bypass the “Comments to the Author” section, enter your conflict of interest statement in the “Confidential to Editor” section, and submit your "Accept" recommendation.

Reviewer #1: All comments have been addressed

Reviewer #2: All comments have been addressed

Reviewer #3: (No Response)

2. Is the manuscript technically sound, and do the data support the conclusions?

Reviewer #1: Yes

Reviewer #2: Yes

Reviewer #3: Yes

3. Has the statistical analysis been performed appropriately and rigorously? 

Reviewer #1: No

Reviewer #2: Yes

Reviewer #3: Yes

4. Have the authors made all data underlying the findings in their manuscript fully available?

Reviewer #1: Yes

Reviewer #2: Yes

Reviewer #3: Yes

5. Is the manuscript presented in an intelligible fashion and written in standard English?

Reviewer #1: Yes

Reviewer #2: Yes

Reviewer #3: Yes

6. Review Comments to the Author

Reviewer #1: All the issues that I previously pointed out were effectively addressed. Indeed, the manuscript changed quite a lot and now the exposition of the ideas is much more clear.

The included sections really help to understand the basis of the different methods, and there are now several references that truly aid in supporting the statements.

Additionally, now the section 3.2.1 really explains in a better way, step by step, all the different methods and the rationale behind them. The table 1 offers a clear overview and categorization of all of them. Now the take-home message (e.g. extremizing he aggregate through the BetaArMean) is much more clear.

The idea of moving the burden of the details of the different methods to Section 6, improves a lot the structure of the manuscript: it is now easier to read.

Some final suggestions: Figure 1, the scale font is completely unreadable (compare the fonts to the ones that you have on Figure 2 which can be perfectly read).

The manuscript is now lengthy, but I believe it fits better what is trying to do: summarizing different methods of mathematically aggregation of decisions from human experts.

Reviewer #2: Reviews:

This is my second-round reviews on this paper. Again, I enjoyed reading this paper. Compared with its previous version, some parts have been further clarified.

This paper provides a comprehensive review of methods on information aggregation, in a scenario of group (experts) decision-making. Although the studies are purely empirical, which may lead to the lack of theoretical background and profundity, the concrete contribution on applications has been impressive and applaudable. Therefore, again, I would like to recommend acceptance of this paper.

For a minor issue, I would suggest the authors revise the abstract. The current abstract contains too much background (motivation, research gap, etc.), and yet, only three sentences (lines 20-24 page 1) regarding your substantial works of this paper. It is better to mention your works, your contribution, and the significance of this study in the abstract.

Reviewer #3: (Comments submitted as a separate file)

7. PLOS authors have the option to publish the peer review history of their article (what does this mean?). If published, this will include your full peer review and any attached files.

Reviewer #1: **Yes: **Rodrigo Ramele

Reviewer #2: **Yes: **Dr. Junyi (Don) CHAI

Reviewer #3: **Yes: **Ville Satopää

---

## [Author Response · Author response to Decision Letter 1]

18 Aug 2021

Please see the response to the reviewers' last comments attached as Review_final.docx

---

## [Editor Report · Decision Letter 2]

19 Aug 2021

Mathematically aggregating experts’ predictions of possible futures

PONE-D-21-01369R2

Dear Dr. Hanea,

We’re pleased to inform you that your manuscript has been judged scientifically suitable for publication and will be formally accepted for publication once it meets all outstanding technical requirements.

Kind regards,

Luca Citi, PhD

Academic Editor

PLOS ONE

---

## [Editor Report · Acceptance letter]

25 Aug 2021

PONE-D-21-01369R2 

Mathematically aggregating experts’ predictions of possible futures 

Dear Dr. Hanea:

I'm pleased to inform you that your manuscript has been deemed suitable for publication in PLOS ONE. Congratulations! Your manuscript is now with our production department. 

Kind regards, 

on behalf of

Dr. Luca Citi 

Academic Editor

PLOS ONE